# Balance Training and Shooting Performance: The Role of Load and the Unstable Surface

**DOI:** 10.3390/jfmk9010017

**Published:** 2024-01-03

**Authors:** Stylianos Kounalakis, Anastasios Karagiannis, Ioannis Kostoulas

**Affiliations:** Faculty of Physical and Cultural Education, Evelpidon Hellenic Army Academy, 16673 Athens, Greece; ankaragiannis@hotmail.com (A.K.); jkost@otenet.gr (I.K.)

**Keywords:** postural sway, external load, military personnel, law enforcement personnel, uneven terrain

## Abstract

Military and law enforcement members’ shooting ability is influenced by their postural balance, which affects their performance and survivability. This study aimed to investigate the effects of a proprioception training program (standing or walking on unstable surfaces) on postural balance and shooting performance. Twenty participants, divided into two groups, completed 60 shots in a shooting simulator while standing, before and after a 4-week proprioception training program. One group (*n* = 10) followed the training program (EXP), while the other group followed the regular military academy program (CON). The shooting was conducted under four conditions: without load on a stable surface, with load on a stable surface, without load on an unstable surface, and with load on an unstable surface. The findings reveal that the training program had a significant impact on the EXP, improving their balance (*p* < 0.01). Additionally, only in the EXP, shooting score and the percentage center of gravity increased (*p* < 0.01) and the stability of the shots, measured by holding time on the target, doubled from 2.2 to 4.5 s (*p* < 0.01). These improvements were more pronounced when participants had a load and/or were on an unstable surface. In conclusion, a proprioception training program could be beneficial for improving postural balance and shooting performance.

## 1. Introduction

Maintaining postural balance is of utmost importance for military and law enforcement personnel, as it enables them to carry out their tasks with remarkable effectiveness. This includes navigating through obstacles, changing direction swiftly, and enhancing overall movement efficiency, particularly when dealing with uneven terrain. The act of shooting is a critical aspect of their job, and postural balance plays a crucial role in determining the extent of body sway and rifle stability [1,2]. Ultimately, this directly affects their occupational performance. A decline in postural balance has been associated with reduced aiming accuracy and shorter target holding time, both of which are related to lower shooting performance [3]. In a military and law enforcement context, personnel’s wellness and ability to survive can be at stake because of the consequences of poor shooting skills. The significance of postural balance in facilitating accurate shooting is reinforced by the observation that proficient rifle shooters show reduced body sway amplitudes during both bipedal standing [4] and shooting [5,6], in contrast to novice shooters.

Multiple factors within military and law enforcement settings, such as the carriage of additional equipment (e.g., protective vests and backpacks) and the negotiation of uneven terrain, can adversely affect the balance of police officers and soldiers [1,5,7,8]. Consequently, this can impact their shooting prowess and overall capacity to endure in combat scenarios. The presence of a backpack and uneven terrain leads to changes in muscle activation patterns required to maintain an upright stance, thus modifying the proprioceptive feedback of the nervous system and postural sway [9,10]. Importantly, the implementation of proprioceptive intervention programs holds the potential to enhance the static postural balance of military and law enforcement personnel on unstable surfaces [11].

Therefore, it is recommended to prioritize the development of postural balance, especially for young soldiers, police officers, and novices engaged in specialized training exercises that require precise body and rifle stabilization [1,5]. Considering the importance of postural balance in shooting performance, it is advisable to promote the use of supplementary proprioception training programs to improve shooters’ postural abilities [1,5,8,12]. The aim of this study was to examine the effects of a proprioception training program on shooting performance in novice military members. Including a backpack carriage and unstable surface in the study was crucial, as these conditions frequently appear and negatively impact postural balance, occupational performance, and wellness during military and law enforcement activities.

## 2. Materials and Methods

### 2.1. Participants

Nineteen male and one female cadets (mean ± SD, age: 18 ± 3, height: 178 ± 7, body mass: 75 ± 8) from the Hellenic Army Academy volunteered to take part in the research. The participants were chosen based on their shooting experience, which should be under 2 years. None of the individuals had prior experience in shooting training or proprioception training programs. Once participants had been verbally and in writing informed about the research process, they provided written consent to take part in it. The Educational Council of the Hellenic Army Academy approved the conduct of the study.

### 2.2. Experimental Procedures

All participants underwent an initial assessment of their anthropometric characteristics. Then, they split into two groups: the control group (*n* = 10) followed the typical school program and the experimental group (*n* = 10) participated in a 4-week training program.

The rifle shooting test, comprising 3 sets of 10 shots from a standing position, was conducted before (Pre) and after (Post) the training program. Assessments were carried out on both stable (onB) and unstable (offB) surfaces. The offB was made using two foaming balance beams. Participants had to stand on them and balance in a bipedal position. The shootings occurred in two additional scenarios. In one scenario, the participants were dressed in combat gear and equipment (L). In the other scenario, they were dressed in athletic attire (noL). Each participant executed the aforementioned procedure (all four conditions) twice before the training program and twice after the training program, on separate days to avoid mental or physical fatigue. It is worth noting that the intra-class correlation between the trials was significant in both groups (ICC = 0.85–0.91, *p* < 0.01), and no significant differences were observed between the two trials. As a result, the shooting data from both days were merged.

All measurements were conducted between 2 p.m. and 5 p.m. at the Hellenic Army Academy, in a temperature-controlled room (25–26 °C) with consistent humidity levels (35–45%). Over the course of the first two visits, participants became familiar with shooting under all conditions (onB-L, offB-L, onB-noL, offB-noL) and with the balance test with or without load. A training program was conducted before and after the assessments. The purpose of the program was to improve postural balance.

### 2.3. Balance Assessment

Postural balance evaluation was performed by a one-foot test on a wooden surface (50 cm wide, 20 cm height) at baseline and immediately after the completion of the 4-week training program. The participant placed one-third of one foot on the board and then raised and bent the other leg at the knee joint and maintained a single-limb stance for as long as possible. The hands were free to perform balancing moves. Three attempts were made for each leg with a break of 30 sec between them. The commencement of each trial was started by the participant at their convenience, with the examiner responsible for timekeeping. If the test participant failed to maintain balance in the first three seconds, the trial was repeated. Balance performance was evaluated by calculating the average time for the 6 attempts.

The participants underwent 12 trials, with 6 attempts conducted while wearing combat gear and equipment, and the other 6 while wearing athletic attire. Preceding the commencement of the test, the examiner conducted a demonstration, and the participant underwent a familiarization trial for each leg, with or without combat gear and equipment.

### 2.4. Training Characteristics

Over a period of four weeks, participants were involved in a training program comprising three sessions per week, with an average duration of 25 min for each session. Previous research conducted over a comparable time frame and duration (4 weeks, 10 min daily) has showed improved stability among military personnel [11]. Eleven training sessions were conducted on average. The duration and the level of difficulty of the program was increased by going from simpler to more complex exercises and by affecting visual information (exercises with closed eyes) in order for the participants to have continuous progress.

For the execution of the intervention program, instruments such as bosu balls, mini trampolines, wobble boards, and balance beams were used. Each training session consisted 3 sets of 4 balance exercises: (a) standing on the mini trampoline with a one leg stance with or without vision (b) balancing with both feet on a wobble board, (c) high knee skipping on bosu, (d) walking on foaming balance beam.

The sessions were conducted as circuit training that lasted for a duration of 14 to 36 min. Between sets, there was a rest period of 2 min. Each exercise lasted 30 s for the first week, 45 s for the next 2 weeks, and 60 s during the last week. The exercise to rest ratio was 1:1. A trainer supervised all sessions, providing initial and continuous instructions and monitoring exercise and resting time.

### 2.5. Analytical Methods and Equipment

Body height measurements were conducted in a standing position with a height gauge (Seca 206; Seca, Hamburg, Germany), and the measurement of body mass was conducted with an electronic precision scale (Seca 813; Seca, Hamburg, Germany).

The combat gear was comprised of combat boots, combat vest with a load equal to the weight of personal ammunition (approximately 5 kg), a rucksack, and an M4A1 rifle. The average weight of the equipment was 22.5 ± 1.7 kg and represented 30% of the participant’s body weight. When selecting equipment and load, the focus was on what is typically used by soldiers and law enforcement recruits [11,13].

A shooting simulator (Noptel ST 2000, Noptel Oy, Oulu, Finland) was used to evaluate shooting ability. The simulator consists of a laser transmitter, an optical glass laser sensitive receiver with an associated paper aiming target, which is a 2.3-cm-diameter circular target located 5 m away. This target simulates a 46-cm-diameter target at 100 m, which is similar to the standard 49-cm-wide, 100-m military silhouette man [14]. The shooting simulator was fixed to the barrel of an M4A1 airsoft carbine rifle. This weapon is used by the armed forces for training and has the same look, feel, and features (dimensions and weight), as the actual U.S. Army M4A1. The airsoft M4A1 carbine rifle with CO_2_ magazines produces realistic noise and feeling during recoil. Participants were given the directive to shoot as quickly and precisely as possible.

The variables analyzed with the Noptel software included: (a) the SCORE (arbitrary units, AU), (b) the percentage Center of Gravity (COG) of shots around a specific point, which is not necessarily the center of the target. The higher the percentage, the higher the COG; (c) the deviation (in cm) of shots in relation to the horizontal (X-dev) and the vertical (Y-dev) axes. The shorter the distance, the smaller the deviation; (d) the holding period (Hold) which is the time in seconds before shooting, during which the laser was held firmly within the holding limits. These limits were determined by three points in which it was considered that the shooting rings were made around them: the center of the target, the center of gravity of the laser at the time of holding (COG not for the shots, but for the course of the shooting laser) and the point of the shot itself; (e) the interval time in seconds between shots (Interval) and (f) Relative Triggering Value (RTV; arbitrary units, AU), shows the ‘cleanness’ of triggering. A smaller RTV means that the motion of the laser during shooting time is smaller compared to the motion of the laser during the holding period. The values of parameters (b) to (f) across the 10 shots were averaged.

### 2.6. Statistical Analyses

Statistical analyses were performed using Statistica 10.0 (StatSoft, Inc., Tulsa, OK, USA). A general linear model (GLM) with repeated measures and a Bonferroni post-hoc comparison was performed [15]. For all analyses, we checked data normality with Shapiro-Wilk test and data sphericity with the Mauchly test. Under no circumstances was there a violation of normality or sphericity, therefore the data did not undergo Log Transformation and the Greenhouse-Geisser correction was not used. We calculated the partial effect size η^2^ as a measure of the magnitude of the effect, which takes values between 0 and 1: 0.02 = small, 0.13 = moderate, and >0.26 = large effect size. Possible correlations between measured variables were explored via calculating the Pearson product-moment correlation coefficient (r) for nondirectional tests (two-tailed). A regression analysis was conducted using the SCORE as the dependent variable and balance time together with HOLD as the independent variables. From regression analysis, R squared and the level of significance were calculated. Data are presented as mean ± SD, or mean ± SE. We set the significance level at *p* ≤ 0.05.

## 3. Results

### 3.1. Balance

In terms of balance, results demonstrated large significant differences in load and no-load conditions. Load affected balance negatively [F(1, 18) = 37, *p* < 0.01, η^2^ = 0.70], but similarly in EXP and CON [interaction effect: F(1, 18) = 0.07, *p* = 0.9].

The training program significantly improved balance for the EXP group. In particular, there was a 77 ± 8% enhancement in the no-load condition and a 75 ± 11% improvement in the with-load condition. These improvements differed significantly from the CON group, which showed no changes [η^2^ = 0.62; interaction effect: F(1, 18) = 26, *p* < 0.01], (Figure 1).

### 3.2. Shooting Performace

In all conditions, the baseline values of shooting variables did not exhibit any significant differences between the experimental and control group [F(1, 18) from 0.25 to 0.70, *p* > 0.05].

In both groups, the shooting SCORE was significantly reduced in load [F(1, 16) = 14, *p* < 0.01] and offB conditions [F(1, 16) = 30, *p* < 0.00] compared to no-load and onB respectively. The training for EXP resulted in a significant improvement in SCORE, which was sustained at a similar level for CON [η^2^ = 0.60; interaction effect: F(1, 18) = 15, *p* = 0.01] (Figure 2). The percentage increase in SCORE was 34%, 40%, 48%, and 66% for onB-noL, onB-L, offB-noL, and offB-L, respectively for EXP, with minor changes ranging from 2% to 6% observed for CON.

Both groups showed a significant decrease in percentage of COG in load [F(1, 16) = 41, *p* < 0.00] and offB conditions [F(1, 16) = 49, *p* < 0.00] compared to no load and onB respectively (Figure 3). A significant improvement in percentage COG was noted post-EXP training compared to CON [η^2^ = 0.58; interaction effect: F(1, 18) = 8.9, *p* < 0.01], as shown in Figure 3. The EXP exhibited a percentage surge of 13%, 22%, 19%, and 28% for onB-noL, onB-L, offB-noL, and offB-L, correspondingly, whereas CON experienced minor changes ranging from −1% to 8%.

Table 1 displays the shooting parameters following the completion of the training program. The load and unstable surface had a negative impact on all shooting parameters in both groups (Table 1 and Appendix A). Nonetheless, the training program only resulted in significant improvements for HOLD, X-dev, and Y-dev. Specifically, the proprioception training program led to a significant improvement in HOLD for EXP in all conditions, whereas no changes were observed for CON. The improvement exhibited a notable increase in conditions involving load (Table 1). Following the training, EXP showed significant reductions in both X-dev and Y-dev, outperforming CON in all conditions. A significant correlation between balance and SCORE (r values ranged from 0.45 to 0.67, *p* < 0.05), as well as with X-dev and Y-dev (r values ranged from 0.42 to 0.58, *p* < 0.05) was noted. Additionally, the combination of balance time and HOLD in a regression model [F(2,13) = 4.13, *p* < 0.05] yields an explanatory power of 40% for SCORE variance.

## 4. Discussion

This study aimed to investigate the influence of a proprioception training program, involving backpack carriage and uneven terrain, on shooting performance. A significant increase in postural balance was observed following a 4-week program focused on proprioception training. This improvement occurred simultaneously with a notable increase in shooting precision, demonstrated by improvements in both the score and the shots’ center of gravity. Furthermore, there was an enhancement in rifle stability, as evidenced by longer holding times and reduced deviation in both the horizontal and vertical axes. Thus, the training program had a more notable effect on situations that encompassed military equipment and unstable surfaces. Consequently, the training program had a significant impact on improving the occupational competence of military personnel. The applicability of these findings extends to law enforcement personnel as well.

### 4.1. Proprioception Training and Postural Balance

During the evaluation of nine commonly performed military tasks across various military services globally, it was determined that postural balance ability represents a crucial skill-related fitness component [16]. The objective behind implementing proprioception training programs is to improve the capacity to maintain postural balance. Based on the literature, achieving the desired outcome entails transmitting accurate and timely joint position information to the neuromuscular control centers of the central nervous system [17]. Research has demonstrated that a proprioceptive intervention program enhanced the postural balance of combat soldiers when standing on unstable surfaces. Specifically, the soldiers’ ability to balance on a bosu ball significantly improved, particularly when doing so with their eyes closed [11]. The analysis of previous data reveals a connection between proprioceptive training and postural balance in both healthy adults and young athletes [18,19]. The current study has corroborated the findings of prior research [11,18,19] and observed that these enhancements, although of lesser magnitude, remain evident when military equipment is integrated. The question of whether the implementation of the proprioception training program with participants wearing the equipment would result in a more pronounced effect on balance enhancement and postural stability in specific military and law enforcement activities, needs to be addressed.

### 4.2. Postural Balance and Shooting Performance

Based on our current knowledge, this study is the first to present the combined effect of external load and an unstable surface on shooting performance, both prior to and following a proprioception training program. Maintaining proper postural balance is crucial for successful shooting, as it directly and indirectly affects rifle stability. Previous studies by Mononen et al. [1] and Era et al. [5] have shown that reduced body sway and minimal rifle movement are correlated with improved shooting accuracy and score. According to Ihalainen et al. [12], maintaining postural balance in the anteroposterior direction is essential for stability while holding the rifle. Improving postural balance in this direction results in enhanced rifle stability. The stability of the rifle, as shown by the act of holding and aiming, plays a crucial role in shooting performance [2,20]. Furthermore, postural balance and gun barrel stability can differentiate between high-scoring and low-scoring shots [6,21].

In line with these findings, we reported an enhanced shooting accuracy and increased rifle stability following a training program designed to enhance balance. The notable correlations between SCORE and balance, coupled with the considerable proportion of the observed variance in SCORE that can be attributed to balance and holding time, amplify these observations. In other studies, the combination of a balance training program with either vibration [22] or respiratory muscle training [23] in rifle athletes demonstrated an enhancement in shooting performance. Nevertheless, these studies were unable to discern the exclusive impact of balance training on the reported outcome.

The unstable ground and the load carried had a negative impact on postural stability and shooting performance in this study. We found that participants who stood on an unstable surface while carrying an external load had the worst shooting scores and a lower percentage of COG. Previous studies have shown the adverse effects of load on soldier and police officer postural sway [8,9] and shooting accuracy [13]. Gil-Cosano et al. [13] documented a 30% decline in shooting score when soldiers wore a backpack (19.5 ± 2.4 kg, equivalent to 24.2 ± 3.4% of their body weight). Additionally, the act of walking on uneven surfaces has a considerable influence on postural stability [24].

It is important to note that the beneficial effect of balance training on shooting was more pronounced when both the rucksack and unstable ground were included in the shooting tests. Specifically, after the training period, the percentage increase in SCORE and percentage COG was twice as high compared to the increase observed in the condition without load and on a stable surface. This could be especially critical in real scenario situations to increase the performance and survivability of the military and law enforcement personnel.

### 4.3. Relevance to Professional Practice and Limitations

Military members and special police officers often have to walk long distances while carrying heavy loads. Additionally, they need to defend themselves before or after these activities, and their shooting performance is crucial for self-defense. However, there is a lack of strategies and interventions aimed at promoting occupational performance for these individuals in the military and law enforcement environments. The current study introduces a novel proprioception training program designed to enhance shooting performance in this particular population.

However, there are certain limitations to consider in this study. These include a small sample size and an imbalance in gender, which could affect how applicable the findings are to the general population. Additionally, it is uncertain whether the effects of the training will last beyond the training sessions, or if ongoing proprioception training is necessary to maintain optimal postural stability and shooting performance. Lastly, the question of how the proprioception training program affects shooting performance when fatigued is still unresolved and remains an important issue.

## 5. Conclusions

According to the study, engaging in a balance training program has been shown to enhance shooting performance. Of particular interest is that this impact is magnified when there is load carriage or when standing on unstable surfaces. Considering the crucial role of postural balance in shooting performance and survival in military and law enforcement environments, it is recommended to prioritize training on postural balance to achieve optimal standing positions and rifle stability. This is particularly essential for novice soldiers and police officers in order to enhance their occupational well-being and performance.

## Figures and Tables

**Figure 1 jfmk-09-00017-f001:**
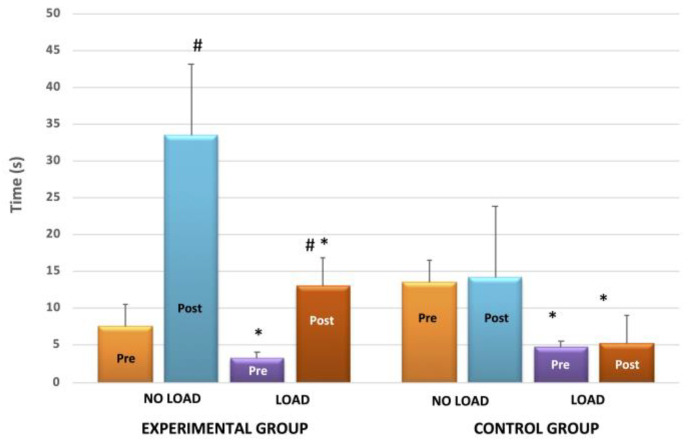
The time in balance test before (PRE) and after (POST) the training period without (NO LOAD) and with load (LOAD), for the experimental (left panel) and control (right panel) groups. The values represent the mean of three trials for each foot, each separated by 30 sec of rest. The plotted values show the mean ± SD. * Significantly different form the respective NO LOAD condition, *p* < 0.01; # Significantly different from the respective PRE condition, *p* < 0.01.

**Figure 2 jfmk-09-00017-f002:**
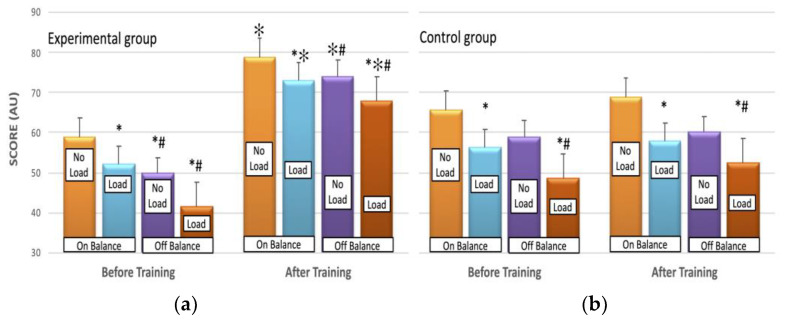
The SCORE before (left panel) and after (right panel) the training period without (No Load) and with load (Load), for the experimental (**a**) and control (**b**) groups. The values represent the mean of six trials of ten shots. The plotted values show the mean ± SE. * significant differences from the respective no-load condition; # significant differences from the respective on balance condition; ✻ significant differences from the respective condition before training, *p* < 0.01.

**Figure 3 jfmk-09-00017-f003:**
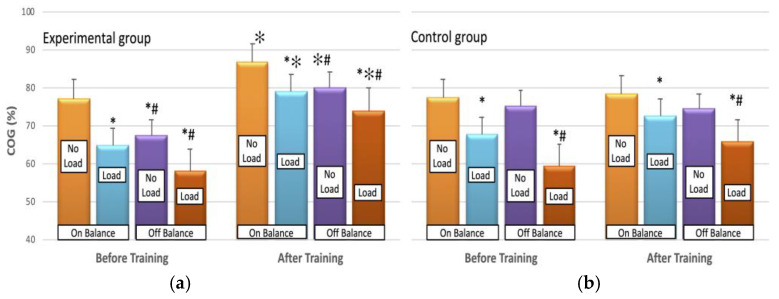
The COG (%) before (left panel) and after (right panel) the training period without (No Load) and with load (Load), for the experimental (**a**) and control (**b**) groups. The values represent the mean of six trials of ten shots. The plotted values show the mean ± SE. * significant differences from the respective no-load condition; # significant differences from the respective on balance condition. ✻ significant differences from the respective condition before training, *p* < 0.01.

**Table 1 jfmk-09-00017-t001:** The changes in shooting parameters after the proprioception training program (values after training minus values before training) for the control and experimental group, on balance on both stable (onB) and unstable (offB) surfaces in either athletic attire (noL) or combat gear and equipment (L).

	onB-noL	onB-L	offB-noL	offB-L
Control Group				
Hold (s)	0.52	0.86	0.28	0.38
Χ-dev (cm)	−0.18	−0.24	0.29	−0.09
Υ-dev (cm)	−0.22	−0.27	0.19	−0.29
Ιnterval (s)	0.79	2.05	1.59	1.31
RTV (AU)	−0.01	0.03	−0.02	−0.03
Experimental Group				
Hold (s)	1.84 #	2.84 #*	1.87 #	2.67 #*
Χ-dev (cm)	−0.46 #	−0.78 #	−0.62 #	−0.62 #
Υ-dev (cm)	−1.00 #	−1.00 #	−0.94 #	−0.95 #
Ιnterval (s)	0.81	−0.28	0.78	1.54
RTV (AU)	−0.07	−0.11	−0.13	−0.07

# significant differences after the training program, *p* < 0.01; * significant differences compared to no load conditions, *p* < 0.01.

## Data Availability

Data are contained within the article and Appendix A.

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
