# Peer review of "Balance Training and Shooting Performance: The Role of Load and the Unstable Surface"

_jfmk, 2024, doi:10.3390/jfmk9010017_

Round 1
Reviewer 1 Report
Comments and Suggestions for Authors
Dear,
Please find my comments attached.
Kind regards

Author Response
For research article “Balance training and shooting performance: the role of load and the unstable surface”
Response to Reviewer 1 Comments
Thank you very much for taking the time to review this manuscript. Please find the detailed responses below and the corresponding revisions/corrections highlighted in the re-submitted file.
Main comments:
An interesting paper, well-structured from a methodological point of view, with comprehensive statistics, figures and discussion (but without Study Limitations and Conclusion sections)
Thank you for the nice comments. The limitations and conclusions are now added in the discussion section.
Point-by-point response to Comments and Suggestions for Authors
Point 1: Although the study had two groups, the attached abstract does not indicate the inclusion of a control group of participants who underwent the typical school program;
The existence of a control group is now stated in the abstract.
Point 2: Authors briefly mentioned previous investigations related to postural balance training, neuromuscular activity, and shooting performance in the Introduction section. However, they have not convinced this reviewer of a gap in the military or sport literature that justifies their study. They need to clearly demonstrate what is known, what is not known, and why the existing gaps justify their study;
The introduction is now re-structured to accomplish these requirements. Please see the introduction section
Point 3: By what criteria were the participants selected for the study? Was the sample normalized?
The criteria were the shooting experience (less than 2 yrs) and no systematic involvement in shooting training or balance training programs. These are now stated in the text.
Point 4: Include Study Limitations section: e.g. The study only measured the effects of the balance training program immediately after its completion. Long-term effects over a more extended period were not assessed;
At the end of paragraphs 4.1 and 4.2, limitations of the study are stated.
Point 5: Include Conclusion section.
The conclusion section is now added in the text.
Reviewer 2 Report
Comments and Suggestions for Authors
Dear Authors,
I hope you are doing very well.
The current content of this paper needs to be largely improved. I hope that my comments can guide you on move your work one step further.
Kind regards

The English writing, and the scientific soundness needs to be carefully revised.
Author Response
Dear reviewer,
We would like to thank you for the constructive comments and suggestions, which have enriched and notably improved the manuscript. We have considered all the comments and revised the manuscript accordingly.
P1L15: changed as you suggested.
P1L16-17: the percentages deleted.
P1L20: “can” changed to “could”.
P1L24: We now restructured the introduction to fulfill these requirements.
P2L53-57: We changed paragraph 2.1 according to your suggestions.
P2L59: these values are added in the previous paragraph.
P2L69: To avoid mental and physical fatigue. This info is now added in the text.
P3L99: The duration of these studies is usually 4-6 weeks. We now added a sentence that states: “Previous research conducted over a similar time frame and frequency has showed enhanced stability among military personnel [11]”.
P3L133: This is the standard equipment used by soldiers. We now added a sentence that states: “When selecting equipment and load, the focus is on what is typically used by soldiers and law enforcement recruits [11,13].”
P3L133: the “%” changed with the word “percentage” throughout the MS.
P4L147-150: We now added one reference to how to treat data in exercise science and reported the normality data check as requested.
P4L158: changed to “Data are presented as mean ± SD, or mean ± SE.”
P4L162: changed as you suggested.
P4L163: the percentage presented here, the absolute values presented in the figures. We have the opinion that the percentages should stay in the text while the absolute values should stay in the figures/Tables.
P5L193: You are right about the dense info in the figures. However, we decided on this interpretation because the reader can see the two main effects: training and condition. Another way was to separate EXP from CON, but the info is not in one figure and the reader has to go back and forth to see the big picture.
P6L216: Changed as you suggested.
P6L217-220: Changed as you suggested.
P6L226-229: Changed as you suggested in order the point to be clear.
P6L234: Refs added.
P7L247: these results are not from our study. The sentence changed to clarify this.
P7L257-272: We now reconstructed the discussion according to your suggestions.
Reviewer 3 Report
Comments and Suggestions for Authors
The paper objective is: The aim of this study was to examine the effects of a balance training program on shooting performance in novice soldiers.
The research authors are recommended to solve the following approaches:
1) The objective described in the abstract section and at the end of the introductory section must be exactly the same. It is recommended to use the research objective described at the end of the introductory section, taking into account its greater accuracy.
2) It is requested to start the discussion section with the research objective and its fulfillment.
3) In the abstract section, it is requested to briefly describe the training program implemented.
4) To establish an experimental investigation with independent samples, it is necessary to previously establish the performance index of each group (establishing the non-existence of significant differences), with a view to determining homogeneity in the variables studied, avoiding effects of foreign variables on the results finals. In the event that the authors cannot establish what was requested, it is recommended to describe the case in a research limitations section.
5) The research uses the Pearson test (Section 2.6), being a parametric statistician, compliance with normality in the data distribution is requested. The authors must specify the normality test used, and whether or not the required standards are met, otherwise another linear correlation indicator must be used.
6) A subsection of Strengths and Limitations of the research is requested, taking into account the small size of the sample, and the great imbalance between genders (only one female subject), as well as the possible negative effects on the study reproducibility.
Comments on the Quality of English LanguageConsult with an English language specialist
Author Response
Dear reviewer,
We would like to thank you for the constructive comments and suggestions, which have enriched and notably improved the manuscript. We have considered all the comments and revised the manuscript accordingly.
1. The objective described in the abstract section and at the end of the introductory section must be exactly the same. It is recommended to use the research objective described at the end of the introductory section, taking into account its greater accuracy.
We now change this according to your suggestion.
2. It is requested to start the discussion section with the research objective and its fulfillment.
Changed as you suggested
3. In the abstract section, it is requested to briefly describe the training program implemented.
We added the basic info of the training program in the abstract.
4. To establish an experimental investigation with independent samples, it is necessary to previously establish the performance index of each group (establishing the non-existence of significant differences), with a view to determining homogeneity in the variables studied, avoiding effects of foreign variables on the results finals. In the event that the authors cannot establish what was requested, it is recommended to describe the case in a research limitations section.
We agree that this is a crucial issue. We check all baseline values in the 2 groups before the training implementation. In section 3.2 we refer to this: “In all conditions, the baseline values of shooting variables did not exhibit any significant differences between the experimental and control group [F(1,18) from 0.25 to 0.70, p > 0.05].”
5. The research uses the Pearson test (Section 2.6), being a parametric statistician, compliance with normality in the data distribution is requested. The authors must specify the normality test used, and whether or not the required standards are met, otherwise another linear correlation indicator must be used.
Thank you for the comment, we added this info in section 2.6
6. A subsection of Strengths and Limitations of the research is requested, taking into account the small size of the sample, and the great imbalance between genders (only one female subject), as well as the possible negative effects on the study reproducibility.
We now added a subsection, as you suggested.
Round 2
Reviewer 1 Report
Comments and Suggestions for Authors
Dear,
please find my comments attached.
Kind regards

Author Response
Done
Thank you for your contribution.
Kind regards
Reviewer 2 Report
Comments and Suggestions for Authors
Dear Authors,
I consider you did a great job revising the paper. However, as I mentioned in the revision before, the entire report of the results needs to be reformulated, and not only the first sentence (as you did).
Kind regards,
Ana

Author Response
Thank you for your comments. Our response is provided below.
P3L110: the sentence changed accordingly
P4L175: In a similar manner, we have performed the same procedure for SCORE and COG. F values were not provided for the remaining parameters, with only significances listed in the table. By following this approach, we ensure the outcomes are kept as uncomplicated as possible.
P7L245: We made a modification to this sentence with the intention of simplifying it.
Reviewer 3 Report
Comments and Suggestions for Authors
The research authors have resolved the allegations issued.
Comments on the Quality of English LanguageCheck with an English language specialist
Author Response
The text has now carefully revised for the English language.
Thank you for your contribution.
Kind regards,
Stylianos Kounalakis
Round 3
Reviewer 2 Report
Comments and Suggestions for Authors
Dear Authors,
Maybe I did not explain myself in the way, I'm sorry. I was not talking about the report of F-values, but about the writing. Every time that you have to report F-values, p-values and/or eta-square values, the report must be as follows: "data showed large significant differences between xxxx and yyy, with xxx performing better than yyy". Thereby, in the writing you start by mention the magnitude of the difference (eta-square). This issue must be corrected throughout the results section.
Kind regards,
Author Response
Dear reviewer,
The necessary modifications have been made to the results as per your feedback (refer to the results section).
Kind regards,
Stylianos Kounalakis